# Parameter-Efficient Masking Networks

**Yue Bai**[1,*]    **Huan Wang**[1,4]    **Xu Ma**[1]    **Yitian Zhang**[1]    **Zhiqiang Tao**[3]    **Yun Fu**[1,2,4]

[1]Department of Electrical and Computer Engineering, Northeastern University
[2]Khoury College of Computer Science, Northeastern University
[3]School of Information, Rochester Institute of Technology
[4]AInnovation Labs, Inc.
Project Homepage: https://yueb17.github.io/PEMN

## Abstract

A deeper network structure generally handles more complicated non-linearity and performs more competitively. Nowadays, advanced network designs often contain a large number of repetitive structures (e.g., Transformer). They empower the network capacity to a new level but also increase the model size inevitably, which is unfriendly to either model restoring or transferring. In this study, we are the first to investigate the representative potential of fixed random weights with limited unique values by learning diverse masks and introduce the Parameter-Efficient Masking Networks (PEMN). It also naturally leads to a new paradigm for model compression to diminish the model size. Concretely, motivated by the repetitive structures in modern neural networks, we utilize one random initialized layer, accompanied with different masks, to convey different feature mappings and represent repetitive network modules. Therefore, the model can be expressed as *one-layer* with a bunch of masks, which significantly reduce the model storage cost. Furthermore, we enhance our strategy by learning masks for a model filled by padding a given random weights vector. In this way, our method can further lower the space complexity, especially for models without many repetitive architectures. We validate the potential of PEMN learning masks on random weights with limited unique values and test its effectiveness for a new compression paradigm based on different network architectures. Code is available at https://github.com/yueb17/PEMN.

## 1 Introduction

Deep neural networks have emerged in several application fields and achieved state-of-the-art performances [8, 16, 29]. Along with the data explosion in this era, huge amount of data gathered to build network models with higher capacity [4, 7, 25]. In addition, researchers also pursue a unified network framework to deal with multi-modal and multi-task problems as a powerful intelligent model [25, 37]. All these trending topics inevitably require even larger and deeper network models to tackle diverse data flows, arising new challenges to compress and transmit models, especially for mobile systems.

Despite the success of recent years with promising task performances, advanced neural networks suffer from their growing size, which causes inconvenience for both model storage and transferring. To reduce the model size of a given network architecture, neural network pruning is a typical technique [22, 20, 12]. Pruning approaches remove redundant weights using designed criteria and the pruning operation can be conducted for both pretrained model (conventional pruning: [13, 12]) and randomly initialized model (pruning at initialization: [21, 32]). Another promising direction is to obtain sparse network by dynamic sparse training [9, 24]. They jointly optimize network architectures and weights to find good sparse networks. Basically, these methods commonly demand regular training, and the final weights are updated by optimization algorithms like SGD automatically.

---

*Corresponding author: `bai.yue@northeastern.edu`

36th Conference on Neural Information Processing Systems (NeurIPS 2022).

Now that the trained weights have such a great representative capacity, one may wonder what is the potential of random and fixed weights or is it possible to achieve the same performance on random weights? If we consider a whole network, the answer is obviously negative as a random network cannot provide informative and distinguishable outputs. However, picking a subnetwork from a random dense network make it possible as feature mapping varies with changes of subnetwork structures. Then, the question has been updated as *what is the representative potential of random and fixed weights with selecting subnetwork structures?* Pioneer work LTH [10] shows the winning ticket exists in random network with good trainability but cannot be used directly without further training. Supermasks [39] enhances the winning ticket and enable it being usable directly. Recent work Popup [26] significantly improves subnetwork capacity from its dense counterpart by learning the masks using backpropagation. Following this insightful perspective, we further ask a question – *what is the maximum representative potential of a set of random weights?* In our work, we first make a thorough exploration of this scientific question to propose our Parameter-Efficient Masking Networks (PEMN). Then, leveraging on the PEMN, we naturally introduce a new network compression paradigm by combining a set of fixed random weights with a corresponding learned mask to represent the whole network.

We start with network architectures which recent popular design style, *i.e.*, building a small-scale encoding module and stacking it to obtain a deep neural network [8, 28, 27]. Based on this point, we naturally propose the *One-layer* strategy by using one module as a prototype and copy its parameter into other repetitive structures. More generally, we further provide two versions: *max-layer padding (MP)* and *random weight padding (RP)* to handle diverse network structures. Specifically, *MP* chooses the layer with the most number of parameters as the prototype and uses first certain parameters of prototype to fill in other layers. *RP* even breaks the constraint of network architecture. It samples a random vector with certain length as the prototype which is copied several times to fill in all layers based on their different lengths. *RP* is architecture-agnostic and can be seen as a most general strategy in our work. Three strategies are from specific to general manner and reduce the number of unique parameters gradually. We first employ these strategies to randomly initialize network. Then, we learn different masks to explore the random weights potential and positively answer the scientific question above. Leveraging on it, we propose a new network compression paradigm by using a set of random weights with a bunch of masks to represent a network model instead of restoring sparse weights for all layers (see Fig. 1). We conduct comprehensive experiments to explore the random weights representative potential and test the model compression performance to validate our paradigm. We summarize our contributions as below:

- We scientifically explore the representative potential of fixed random weights with limited unique values and introduce our Parameter-Efficient Masking Networks (PEMN). It leverages on learning different masks to represent different feature mappings.

- A novel network compression paradigm is naturally proposed by fully utilizing the representative capacity of random weights. We represent and restore a network based on a given random vector with a bunch of masks instead of retaining all the sparse weights.

- Extensive experimental results explore the random weights potential by using our PEMN and test the compression performance of our new paradigm. We expect our work can inspire more interesting explorations in this direction.

## 2 Related Works

### 2.1 Sparse Network Training

Our work is related to sparse network training. Conventional pruning techniques finetune the pruned network from pretrained models [12, 13] with various pruning criteria for different applications [22, 14, 15, 33, 31, 35, 36]. Instead of pruning a pretrained model, pruning at initialization [32] approaches attempt to find winning ticket from the random weight. Gradient information is considered to build pruning criteria in [21, 30]. Different from pruning methods above, sparse network training also can be conducted in a dynamic fashion. To name a few, Rigging the Lottery [9] edits the network connections and jointly updates the learnable weights. Dynamic Sparse Reparameterization [24] modifies the parameter budget among the whole network dynamically. Sparse Networks from Scratch [6] proposes a momentum based approach to adaptively grow weights and empirically verifies its effectiveness.

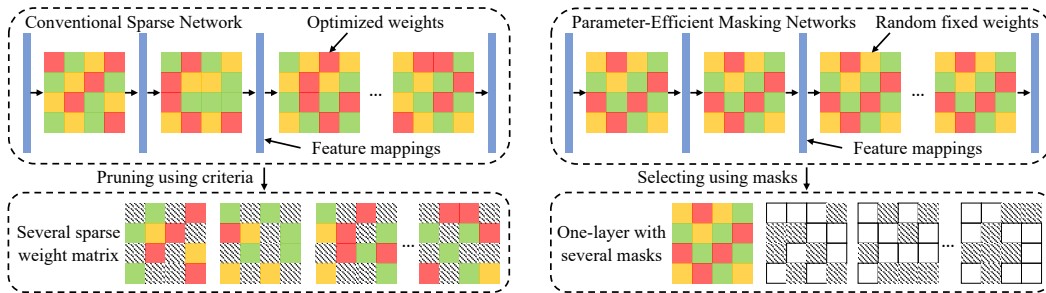

Figure 1: Comparison of different ways to represent a neural network. Different features mappings are shown as blue rectangles. Squares with different color patches inside serve as parameters of different layers. Left is the conventional fashion where weights are optimized and sparse structures are decided by certain criteria. Right is our PEMN to represent a network where the prototype weights are fixed and repetitively used to fill in the whole network and different masks are learned to deliver different feature mappings. Following this line, we explore the representative potential of random weights and propose a novel paradigm to achieve model compression by combining a set of random weights and a bunch of masks.

Most of the sparse network training achieve the network sparsity by keeping necessary weights and removing others, which reduces the cost of model storage and transferring. In our work, we propose a novel model compression paradigm by leveraging the representative potential of random weights accompanied with subnetwork selection.

## 2.2 Random Network Selection

Our work inherits the research line of exploring the representative capacity of random network. The potential of randomly initialized network is pioneeringly explored by the Lottery Ticket Hypothesis [10], and further investigated by [23, 2, 34]. It articulates that there exists a winning ticket subnetwork in a random dense network. This subnetwork can be trained in isolation and achieves comparable results with its dense counterpart. Moreover, the potential of the winning ticket is further explored in Supermasks [39]. It surprisingly discovers the subnetwork can be identified from dense network to obtain reasonable performance without training. It extends and proves the potential of subnetwork from good trainability to being used directly. More recently, the representative capacity of subnetworks is enhanced by Popup algorithm proposed by [26]. Based on random dense initialization, the learnable mask is optimized to obtain subnetwork with promising results. Instead of considering network with random weights, the network with the same shared parameters can also delivery representative capacity to some extent, which is investigated by Weight Agnostic Neural Network [11] and also inspires this research direction. We are highly motivated by these researches to validate how is the representative potential of random weights with limited unique values by learning various masks.

## 2.3 Weight Sharing

Our study is also related to several recent works about weight sharing. This strategy has been explored and analyzed in convolutional neural networks for their efficiency [17, 38]. In addition, several works are also proposed for efficient transformer architecture using weight sharing strategy [19, 5, 1]. There are two main differences between these works and our study: 1) They follow the regular optimization strategy to learn the weight in a recurrent fashion, which is closer to the recurrent neural network. Our work follows a different setting. We use fixed repetitive random weights to fill in the whole network and employs different masks to represent different feature mappings; 2) They mainly conduct cross-layer weight sharing for repetitive transformer structure. In our work, we explore the potential of random weight vector with limited length as much smaller repetitive granularity to fill in the whole network, which is more challenging than cross-layer sharing strategy.

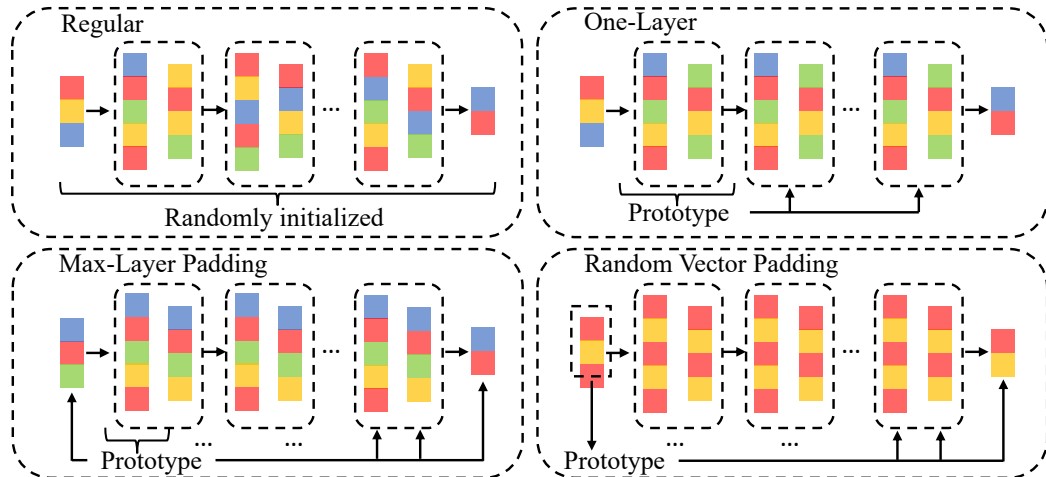

Figure 2: Illustrations of different strategies in PEMN to represent network structures. Compared with regular fashion where all parameters are randomly initialized, we provide three parameter-efficient strategies, *One-layer*, *Max-layer padding (MP)*, and *Random vector padding (RP)*, to fully explore the representative capacity of random weights.

## 3 Parameter-Efficient Masking Networks

### 3.1 Instinctive Motivation

Overparameterized randomly initialized neural network benefits network optimization to get higher performance. Inevitably, the trained network contains redundant parameters but can be further compressed, which defines the conventional neural network pruning. On the other side, the network redundancy also ensures a large random network contains a huge number of possible subnetworks, thus, carefully selecting a specific subnetwork should obtain promising performances. This point of view has been proved by [26, 39]. These works demonstrate the representative potential of certain subset combinations of a given random weights. Following this lane, we naturally ask a question: *what is the maximum representative potential of a set of random weights?* or in another word: *can we use random weights with limited unique values to represent a usable network?* We answer this question as positive and introduce our Parameter-Efficient Masking Networks (PEMN). Moreover, leveraging on 1) compared with trained network where the weight values cannot be predicted, we can pre-access the random weights before we select the subnetwork; 2) selected subnetwork can be efficiently represented by a bunch of masks, we can extremely reduce the network storage size and establish a new paradigm for network compression.

### 3.2 Sparse Selection

We follow [26] to conduct the sparse network selection. We start from a randomly initialized neural network consisting of $L$ layers. For each $l \in \{1, 2, ..., L\}$, it has

$$\mathcal{I}_{l+1} = \sigma(\mathcal{F}[\mathcal{I}_l; w_l]), \tag{1}$$

where $\mathcal{I}_l$ and $\mathcal{I}_{l+1}$ are the input and output of layer $l$. $\sigma$ is the activation. $\mathcal{F}$ represents the encoding layer such as convolutional or linear layer with parameter $w_l = \{w_l^1, w_l^2, ..., w_l^{d_l}\}$, where $d_l$ is the parameter dimension of layer $l$. To perform the sparse selection, all the weights $w = \{w_1, w_2, ..., w_L\}$ are fixed and denoted as $\widetilde{w}$. To pick the fixed weights for subnetwork, each weight $w_l^j$ is assigned a learnable element-wise score $s_l^j$ to indicate its importance in the network. The Eq. 1 is rewrited as

$$\mathcal{I}_{l+1} = \sigma(\mathcal{F}[\mathcal{I}_l; w_l \odot h(s_l)]), \tag{2}$$

where $s_l = \{s_l^1, s_l^2, ..., s_l^{d_l}\}$ is the score vector and $h(\cdot)$ is the indicator function to create the mask. It outputs 1 when the value of $s_l^j$ belongs to the top K% highest scores and outputs 0 for others, where K is predefined sparse selection ratio. Through optimizing $s$ with fixed $w$, a subset of original dense

weights is finally selected. Since $h(\cdot)$ is a non-derivable function, the gradient of each $s_l^j$ cannot be obtained directly. The straight-through gradient estimator [3] is applied to treat $h(\cdot)$ as identity function during gradient backwards pass. Formally, the gradient of $s$ is approximately computed as

$$\widetilde{g}(s_l^j) = \frac{\partial \mathcal{L}}{\partial \widetilde{\mathcal{I}}_{l+1}} \frac{\partial \widetilde{\mathcal{I}}_{l+1}}{\partial s_l^j} \approx \frac{\partial \mathcal{L}}{\partial \mathcal{I}_{l+1}} \frac{\partial \mathcal{I}_{l+1}}{\partial s_l^j}, \tag{3}$$

where $\widetilde{\mathcal{I}}_{l+1} = \sigma(\mathcal{F}[\mathcal{I}_l; w_l \odot s_l])$, which is applied estimation. $\widetilde{g}(s_l^j)$ is approximately estimated gradient of weight score $s_l^j$. In this way, the dense network is randomly initialized but fixed, but one of its subnetwork can be selected using Backpropagation. In our work, we name this optimization process as *sparse selection*.

## 3.3 Parameter-Efficient Strategy

Following the logic of parameter-efficient exploration from specific to general scenario, PEMN utilizes three strategies to construct the whole network based on given random weights: 1) *One-layer* (Sec. 3.3.1), 2) *Max-layer padding (MP)* (Sec. 3.4), and 3) *Random vector padding (RP)* (Sec. 3.4), which are detailedly introduced below.

### 3.3.1 One-Layer

*Sparse selection* initializes a dense network as a pool to pick certain weights. It provides a novel direction to find admirable subnetwork without pre-training and pruning. However, with increasing of network scales, the cost of restoring and transferring a neural network grows rapidly. Noticed that more popular network structures follow a similar design style: *proposing a well-designed modeling block and stacking it several times to boost network capacity*, we are inspired to explore the feasibility of finding subnetworks by iteratively selecting different masks in a series of repetitive modules. Formally, a $L$-layer randomly initialized network $\mathcal{N}_L$ can be represented as a series of parameters:

$$\mathcal{N}_L : w = [w_1, w_2, ..., w_L]; w_l \in \mathbb{R}_l, \tag{4}$$

where $l \in \{1, 2, ..., L\}$ and $w_l$ is used for various layers. $\mathbb{R}_l$ denotes different parameter spaces (e.g., $\mathbb{R}_l^{I \times O}$ for linear, $\mathbb{R}_l^{N \times H \times W}$ for CNN layer, where $I/O$ are input/output dimensions and $N/H/W$ are CNN kernel dimensions). From shallow to deep

---

**Algorithm 1** One-Layer

1: **Input:** A random network with L-layer: $w = \{w_1, w_2, ..., w_L\}$
2: **Output:** A L-layer network filled by P prototype layers with parameters: $w^* = \{w_{pro^1}, w_{pro^2}, ..., w_{pro^P}\}$
3: Randomly initialize layers from 1 to L
4: Record parameter space dimensions: $\{\mathbb{R}_1, \mathbb{R}_2, ..., \mathbb{R}_L\}$
5: Initialize a prototype layers list: $List_{pro} = []$
6: **for** $l$ in $1, 2, ..., L$ **do**
7:     **if** $\forall \mathbb{R}_{pro^p} \in List_{pro} \neq \mathbb{R}_l$ **then**
8:         Append $w_l$ into $List_{pro}$
9:     **else**
10:         Find $w_{pro^p}$, where $\mathbb{R}_{pro^p} = \mathbb{R}_l$
11:         Replace $w_l$ with $w_{pro^p}$
12:     **end if**
13: **end for**
14: Return updated $w$ as $w^*$

---

layer, we first sample the *prototype* layers for the whole network with unique parameter spaces, represented as $w_{pro} = [w_{pro^1}, w_{pro^2}, ..., w_{pro^P}]$. For each $w_{pro^p}$, we use its parameters to replace its *target* layers $w_{tar^p}$ which share the same parameter space:

$$w_{tar_t^p} \leftarrow w_{pro^p}; \mathbb{R}_{tar_t^p} = \mathbb{R}_{pro^p}, t \in \{1, 2, ..., T^p\}, \tag{5}$$

where $T^p$ is the number of target layers of prototype $p$. The whole network filled by several layers with unique weight size and Eq. 4 can be rewrited as

$$\mathcal{N}_L : w^* = [\overbrace{w_{pro^1}, ..., w_{pro^2}, ..., w_{pro^p}, ..., w_{pro^P}}^{T^1 + T^2 + ... + T^P}]; \sum T^p = L. \tag{6}$$

We take a simple 5-layer MLP to clarify this operation: its dimensions are $[512, 100, 100, 100, 10]$ with 4 weight matrices $w_1 \in \mathbb{R}^{512 \times 100}$, $w_2 \in \mathbb{R}^{100 \times 100}$, $w_3 \in \mathbb{R}^{100 \times 100}$, and $w_4 \in \mathbb{R}^{100 \times 10}$. In this case, $w_1$, $w_2$ and $w_4$ are three prototype layers. $w_2$ has two target layers $w_3$ and itself. $w_1/w_4$ only has itself as target layer. The general algorithm is summarized in Alg. 1.

In this way, any repetitive modules in a given network structure can be represented by one bunch of random weights. Using the *sparse selection* strategy, we iteratively pick subnetworks in the same set of random weights to obtain diverse feature mappings. Therefore, the cost to represent the network significantly reduces, especially for deep network with many repetitive blocks. In other words, one random layer with different masks can represent the majority of a complete network structure, which is named as *one-layer*.

## 3.4 Random Weights Padding

*One-layer* strategy efficiently handles networks with many repetitive modules. The majority of the whole network can be compressed into one random layer with a set of masks. However, in real-world applications, various network architectures may not follow a tidy pattern resulting in different shapes for different layers. Hence, the *one-layer* is not flexible enough to efficiently represent such networks. To handle it, we naturally propose a enhanced strategy, *Random Weights Padding*. It consists of two versions, *Max-layer padding* and *Random vector padding*. We first formally rewrite Eq. 4 as

$$\mathcal{N}_L : w = [w_1, w_2, ..., w_L]; w_l \in \mathbb{R}^{d_l}, l \in \{1, 2, ..., L\}, \tag{7}$$

where $w_l$ is flatten into a vector with dimension $d_l$ (e.g., $d_l = I \times O$ for linear layer and $d_l = N \times H \times W$ for CNN layer).

**Max-Layer Padding (MP).** It chooses the layer $w_m$ as the prototype where $d_m = max([d_1, d_2, ..., d_L])$ is the highest dimension. All other layers in $\mathcal{N}_L$ have fewer parameters than $w_m$. We keep the prototype as it is and simply pick the first $d_l$ parameters from $w_m$ to replace the parameters in $w_l$, which is described by

$$w_l \leftarrow w_m[: d_l]; l \in \{1, 2, ..., L\}. \tag{8}$$

**Random Vector Padding (RP).** Instead of picking a complete layer as prototype, RP further reduces the granularity of random prototype from a layer to a random weights vector with a relatively short length. We let $v_{pro} \in \mathbb{R}^{d_v}$ as the random weights vector with length $d_v$. For each layer $l$, we repeat $v_{pro}$ several times to reach the length of $w_l$. It can be formally described as

$$w_l \leftarrow [\overbrace{v_{pro}, ...}^{d_l}]; l \in \{1, 2, ..., L\}. \tag{9}$$

After the padding operation, weights in Eq. 7 are reshaped back into the format of Eq. 4 to perform as a network. These two padding strategies are summarized in Alg. 2 and Alg. 3.

In the series of *one-layer*, *MP*, and *RP*, based on *sparse selection*, we explore using fewer unique weights to represent the whole network. Leveraging on the property of the fixed weight values, the cost of delegating a network keeps decreasing by using a random vector with a bunch of masks. By this mean, we fully explore the representative capacity of random weights with limited unique values. Furthermore, a novel model compression paradigm can be correspondingly established by restoring a set of random weights with different masks. Our three strategies compared to regular network setting are shown in Fig. 2.

---

**Algorithm 2** Max-Layer Padding

1: **Input:** A L-layer random network with parameters: $w = \{w_1, w_2, ..., w_L\}$
2: **Output:** A L-layer network with MP: $w^* = \{w_m[: d_1], w_m[: d_2], ..., w_m[: d_L]\}$
3: Randomly initialize layers from 1 to L
4: Find the layer $w_m$ with the maximum weight dimension $d_m$ among all layers $w_l, l = \{1, 2, ..., L\}$
5: **for** $l$ in $1, 2, ..., L$ **do**
6:   Replace $w_l$ with the first $d_l$ values in $w_m$ given by $w_m[: d_l]$
7: **end for**
8: Return updated $w$ as $w^*$

---

**Algorithm 3** Random Vector Padding

1: **Input:** A L-layer random network with parameters: $w = \{w_1, w_2, ..., w_L\}$
2: **Output:** A L-layer network with RP: $w^* = \{[\overbrace{v_{pro}, ...}^{d_1}], [\overbrace{v_{pro}, ...}^{d_2}], ..., [\overbrace{v_{pro}, ...}^{d_L}]\}$
3: Randomly initialize layers from 1 to L
4: Randomly initialize a weights vector $v_{pro} \in \mathbb{R}^{d_v}$ with $d_v$ dimension
5: **for** $l$ in $1, 2, ..., L$ **do**
6:   Repeat $v_{pro}$ until reaching the length $d_l$
7:   Replace $w_l$ with the repeated vector $v_{pro}$
8: **end for**
9: Return updated $w$ as $w^*$

---

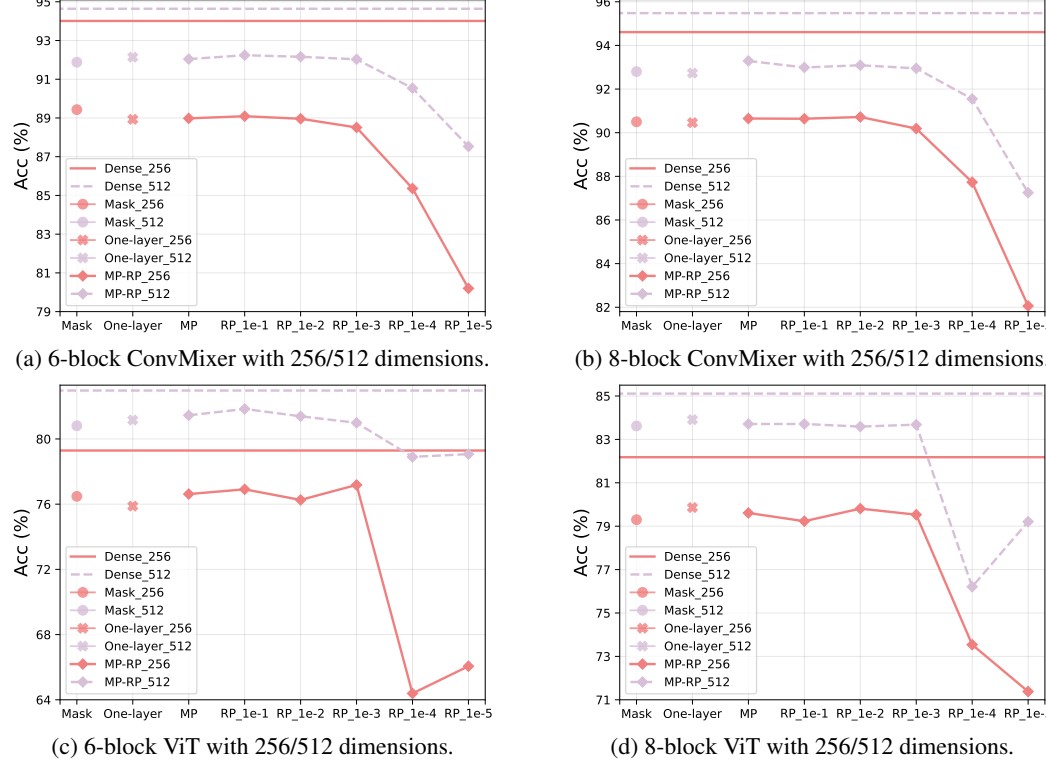

(a) 6-block ConvMixer with 256/512 dimensions.

(b) 8-block ConvMixer with 256/512 dimensions.

(c) 6-block ViT with 256/512 dimensions.

(d) 8-block ViT with 256/512 dimensions.

Figure 3: Performances of ConvMixer and ViT backbones on CIFAR10 dataset with different model hyperparameters. Y-axis represent the test accuracy and X-axis denotes different network parameter settings. *Dense* means the model is trained in regular fashion. *Mask* is the sparse selection strategy. *One-layer*, MP, and *RP* are our strategies. The decimal after *RP* means the number of unique parameters compared with *MP*. From *Mask* to *RP 1e-5*, the unique values of network decrease. Different experimental settings illustrate the representative potential of random weights.

# 4    Experiments

Our experiments conduct empirically validations on two aspects of our interests. Firstly, we validate how large is the representative potential of random weights with limited unique values to test the effectiveness of our proposed Parameter-Efficient Masking Networks (PEMN). Secondly, leveraging on the pre-accessibility of random weights and lightweight storage cost of binary mask, it is promising to establish a new model compression paradigm.

## 4.1    Preparation

We comprehensively use several classic or recently popular backbones for image classification task to conduct general validations. Backbones include ResNet32, ResNet56 [16], ConvMixer [28], and ViT [8]. We use CIFAR10 and CIFAR100 datasets [18] for our experiments.

## 4.2    Representative Random Weights in PEMN

We first explore the representative potential of random weights in PEMN based on our proposed strategies, *One-Layer*, *Max-Layer Padding (MP)*, and *Random Vector Padding (RP)*. We use a CNN based architecture Convmixer [28] and a MLP based model ViT [8] to conduct our experiments on CIFAR10 dataset [18].

In Fig. 3, we show 8 pairs of experiments based on 2 backbones (ConvMixer, ViT) using 2 depth numbers (6, 8) and 2 hidden dimensions (256, 512). Each pair includes a dense network performance and a series of results obtained by *sparse selection* with different random weighting strategies. Specifically, *Mask* learns the mask on the randomly initialized network. *One-layer*, *MP*, and *RP* represent our proposed strategies. To simplify the comparison, we use a rate number after *RP* to show how many unique parameters used in *RP* compared with *MP*. From the left *Mask* to right

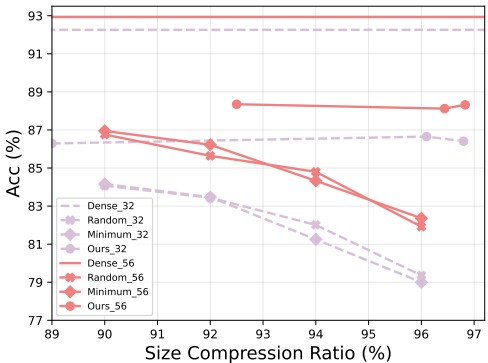 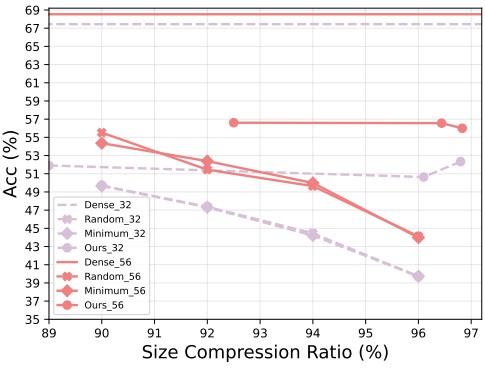

(a) ResNet32/ResNet56 on CIFAR10 dataset.  (b) Resnet32/ResNet56 on CIFAR100 dataset.

Figure 4: Compression performance validation on CIFAR10/CIFAR100 datasets on ResNet32/ResNet56 backbones. Y-axis denotes the test accuracy. X-axis means the network size compression ratio. Different colors represent different network architectures. The straight lines on the top are performance of dense model with regular training. Lines with different symbol shapes denote different settings. For ResNet, our three points are based on *MP*, *RP 1e-1*, and *RP 1e-2*, respectively. This pair of figures show that our proposed paradigm achieves admirable compression performance compared with baselines. In very high compression ratios, we can still maintain the test accuracy.

*RP 1e-5*, the number of unique parameters gradually decreases. Different settings show the similar patterns concluded as: 1) Compared with regular trained dense model, sparse selection approach generally obtains promising results, even if with a performance drop caused by the constraint of fixing all the parameters; 2) From left to right on X-axis, the performance gradually drops. This is caused by the decreasing number of unique values in network, which makes network has less representative capacity; 3) However, performance drop arises when the number of unique parameters is extremely low (e.g., *RP 1e-4*, *RP 1e-5*). The results remain stable for the most of random weights strategies; 4) Larger depth and hidden dimension boost the model capacity for different configurations. The performance drop of random weights strategies from their dense counterpart is also decreased. In addition, ViT shows some unstable fluctuation when fewer unique parameter, compared with ConvMixer with relatively stable patterns. This may caused by the difficulty of training MLP based network itself and will not affect our main conclusions.

The performance stability shown above illustrates the network representative capacity can be realized not only by overparameterizing the model, but also carefully picking different combinations of random parameters with limited unique values. In this way, the effectiveness of our PEMN is validated and we can represent a network using a random parameters prototype with different learned masks, instead of typically restoring all the different parameters. This property inspires us to introduce a new model compression paradigm proposed in following section.

## 4.3 A New Model Compression Paradigm

Practically, our work proposes a new network compression paradigm based on a group of random weights with different masks. We first elaborate the network compression and storage processes to clarify our advantages then report the empirical results.

### 4.3.1 Sparse Network Storage

Previous works aim to remove redundant weights (e.g., unstructured pruning) among different layers. The trivial weights are set to zero based on different criteria. The ratio of zero-weight in the whole network is regarded as sparsity ratio. Different approaches are compared based on their final test accuracy with a given sparsity ratio. Different from this conventional fashion which restoring sparse trained weights, we instead use fixed random weights with different masks to represent a network. To compare these two paradigm, we calculate the required storage size as an integrated measurement.

Assuming we have a trained network with $p$ as parameter numbers and $r$ as sparsity ratio. Due to its sparsity, we only need to restore the non-zero weight values accompanied with their position [12] denoted as a binary mask. The storage cost can be separated into two parts, $C_w$ for weight values and

$C_m$ for mask given by

$$C = C_w + C_m. \tag{10}$$

For conventional setting, $C_w$ restores the values of kept sparse weights which is $p \cdot (1 - r)$ for the whole network. It needs to be restore in float format. $C_m$ restores sparse positions of these weights, which can be restored into compressed sparse column (CSC) or compressed sparse row (CSR) formats with cost around $2p \cdot (1 - r)$ [12]. In our new paradigm, $C_w$ records the values of the given random weights. For example, the *one-layer* requires to record all weights of non-repetitive layers and one prototype weights of all repetitive layers. *MP* requires to keep the values of layer with the largest number of parameters. *RP* only requires to restore the values of a random vector with given length. $C_m$ is also for the sparse positions to record the selected subnetwork.

### 4.3.2 Compression Performance Validation

We test the compression performance on CI-FAR10 and CIFAR100 datasets using ConvMixer with 6/8 depths, ResNet32, and ResNet56 backbones. The compression ratio is based on the storage size as we discussed instead of the conventional pruning ratio. Since we propose a new strategy to compress network, we involve two sparse network training baselines in our experiments. Specifically, we train a sparse network from scratch by removing random weight and minimum magnitude weights. For compression ratio, we set four settings for baselines: 90%, 92%, 94%, and 96%. We directly refer to settings, *MP*, *RP* from Sec. 4.2 for our paradigm. Their compression ratio is calculated using the same measurement as baselines.

In Fig. 4, we show 4 pairs of comparison based on 2 backbones (ResNet32/ResNet56) and 2 datasets (CIFAR10/CIFAR100). Each pair includes dense model, 2 baselines with 4 compression ratios, and our results in 3 ratios. Two baselines are sparse network training by pruning random weights and minimum magnitude weights, respectively. For convenience, we do

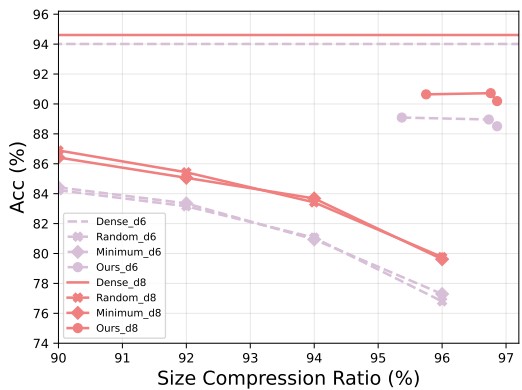

Figure 5: Compression performance validation on CIFAR10 dataset on ConvMixer backbone. Y-axis is the test accuracy. X-axis means compression ratio. Two pairs of comparisons are for different depths shown in different colors. Straight line on the top is the dense model performance. Curves in different symbols represent baselines and our method. For ConvMixer, our three points are based on *RP 1e-1*, *RP 1e-2*, and *RP 1e-3*, respectively.

not follow exactly the same compression ratios as baseline but directly use settings from Sec.4.2. For ResNet, we use *MP*, *RP 1e-1*, and *RP 1e-2*. Their corresponding compression ratios are computed as shown in figures. Experiments based on different networks and datasets show the similar conclusions summarized as: 1) our method outperforms the baselines by a significant margin, with even higher compression ratio; 2); Compared with conventional sparse network training where compressed model performance decreases obviously along with increasing compression ratio, our method is relatively robust to the compressed model size; 3) If we compare cross different models, we find compressed small model by our method even performs better than baselines using larger model; 4) Network scale affects the compression performance, compared with ResNet32, ResNet56 basically contains more parameters and performance drop between compressed network with its dense counterpart is relatively small. In Fig. 5, we show compression performance on CIFAR10 dataset using ConvMixer with depth 6 and 8. The settings are basically similar to Fig. 4. We can also draw the similar conclusions: 1) Our method outperforms the baselines on ConvMixer with different depths; 2) Our method compresses network into lower size but maintains higher performance.

As a summary, in Sec. 4, our experiments can be separated into two parts. Firstly, to validate our PEMN, we investigate the representative potential of random weights which are used to fill in the complete network structure using different proposed strategies. Secondly, the promising conclusion (Sec. 4.2) for this investigation naturally leads to the newly proposed network compression paradigm. Different from conventional fashion restoring sparse weights, we instead restore the fixed random weights and different masks. Empirically, we validate the effectiveness of our new paradigm for

network compression. Our experiments involve diverse network architectures to demonstrate the proposed paradigm can be generalized into different network designs.

## 5 Discussion and Conclusion

**Discussion** We summarize our intuitive logic and potential research direction in the future. Our fundamental insight is motivated by Supermasks [39] and Popup [26] showing random network encodes informative pattern by selecting subnetworks. They inspire us to understand neural network in a decoupled perspective: the informative output is delivered by certain weight-structure combination. Even if weights are fixed, the flexibility of learnable masks still provides promising capacity to represent diverse semantic information. We are the first to fully explore the representative potential of random weights, and practically, a new network compression paradigm is naturally established. We further discuss some research directions in the future following this study. Firstly, compared with conventional approaches need to record learned weights, our paradigm records random weights which can be pre-accessed, can this property be used for improve the model security? In addition, leveraging on the property that repetitive random weights existing in networks for our strategies, is it possible to specifically design hardware deployment configurations to achieve further compression or acceleration? Moreover, our PEMN is based on random initialized weights but cannot be directly deployed on pretrained models, which are more powerful these days. How to further improve our strategy on large-scale pretrained models is another interesting point to explore. We leave these topics in our future work.

**Conclusion** We first explore the maximum representative potential of a set of fixed random weights, which leverages different learned masks to obtain different feature mappings. Correspondingly, we introduce our proposed Parameter-Efficient Masking Networks (PEMN). Specifically, we naturally propose three strategies, *one-layer*, *max-layer padding (MP)*, and *random vector padding (RP)*, to fill in a complete network with given random weights. We find that a neural network with even limited unique parameters can achieve promising performance. It shows that parameters with fewer unique values have great representative potential achieved by learning different masks. Therefore, we can represent a complete network by combining a set of random weights with different masks. Inspired by this observation, we propose a novel network compression paradigm. Compared with traditional approaches, our paradigm can restore and transfer a network by only keeping a random vector with masks, instead of recording sparse weights for the whole network. Since the cost of restoring a mask is significantly lower than weight, we can achieve admirable compression performance. We conduct comprehensive experiments based on several popular network architectures to explore the random weights potential for PEMN and test the compression performance of our new paradigm. We expect our work can inspire further researches for both exploring network representative potential and network compression.

## Acknowledgments and Disclosure of Funding

We thank the anonymous NeurIPS reviewers for giving us valuable and constructive suggestions to improve our paper. This work is supported by Northeastern University and finished by Yue working in SmileLab as a research assistant. There are no competing interests to disclose.

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
