# Supplementary Material for Parameter-Efficient Masking Networks

**Yue Bai**[1,*]   **Huan Wang**[1,4]   **Xu Ma**[1]   **Yitian Zhang**[1]   **Zhiqiang Tao**[3]   **Yun Fu**[1,2,4]

[1]Department of Electrical and Computer Engineering, Northeastern University
[2]Khoury College of Computer Science, Northeastern University
[3]School of Information, Rochester Institute of Technology
[4]AInnovation Labs, Inc.
Project Homepage: https://yueb17.github.io/PEMN

## A.   Implementation Details

For all the backbones used in our experiments, we follow their default training settings. For ConvMixer [10], we use AdamW [8] optimizer with triangular learning rate scheduler. We set the maximum learning rate as 0.05 with weight decay as 0.005. We set batch size as 512 and the number of total epochs is 100. We use different configurations for hidden dimension (256/512) and depth (6/8) in our experiments section. For ViT [2], we use Adam [5] optimizer with cosine learning rate scheduler. We set the maximum learning rate as 0.0001. We set batch size as 256 and the number of total epochs as 200. We use different configurations for hidden dimension (256/512) and depth (6/8) in our experiments section. For ResNet [4], we follow the implementation configurations in Popup [9]. Specifically, we use SGD optimizer with maximum learning rate 0.1 and cosine scheduler. The weight decay and momentum are set as 0.0005 and 0.9. We train 100 epochs with 256 batch size. For the sparse selection to pick the subnetworks, we follow the optimal choice in Popup and set it as 0.5 for our experiments. And we use kaiming uniform and kaiming normal [3] to initialize the scores and random weights, respectively.

## B.   Limitations and Potential Negative Societal Impacts

Firstly, our study focuses on exploring the random weight potential and representing a neural network with small storage cost. However, it cannot further help for accelerating the training and inference. This is a good point for us to further explore. Secondly, our proposed network compression strategy leverages on the representative potential of random weights with different masks, which benefits to reducing storage cost. However, following this strategy, it is hard to take advantages of powerful pretrained model these days. How to tailor our insight to large-scale pretrained model is another good point to explore. To the best of our knowledge, our study has no potential negative societal impacts.

## C.   More Experimental Evaluations

We further add experiments for CIFAR100 [6] and Tiny-ImageNet [7] datasets using ConvMixer as backbone to test our new network compression paradigm. The results are shown in Fig. 1. We use 8 and 6 as depth number for CIFAR100 and Tiny-ImageNet datasets, respectively and other settings follow the same mentioned above. The results in figure show that our compression strategy outperforms the baseline methods and validate the effectiveness of our propose network compression paradigm. We also conduct experiments on large-scale ImageNet [1] dataset using ResNet50 [4] as backbone. For simplicity, we customize two ImageNet subset for convenient evaluation. We sample

---

*Corresponding author: `bai.yue@northeastern.edu`

36th Conference on Neural Information Processing Systems (NeurIPS 2022).

100/200 classes from original ImageNet to construct our subsets. We compare our compression strategy with magnitude pruning baseline and results are shown in Fig. 2. The results of our method are promising and demonstrate its effectiveness on challenging ImageNet dataset.

## D.  The Results of Repeated Experiments

We further supplement the repeated experimental results. We supplement the main results from Fig. 3 and Fig. 4 in the *Experiments* section. We repeated each experiment three times and report mean and std. In Tab. 1, Tab. 2, Tab. 3, Tab. 4, we provide the repeated results for Fig. 3. According to these results, we make several conclusions: 1) Our experimental performance are generally consistent across different datasets and different settings. The supplemented results follow the accuracy patterns and support the conclusions provided in our draft; 2) ConvMixer is more stable than ViT backbone across different settings; 3) Overall, along with the decreasing number of unique values in the network (from the left column to the right column of tables), the performance variations increase correspondingly. The limited unique weight values decreases the stability of the network. In Tab. 5, Tab. 6, we provide the repeated results for Fig. 4. The first four columns show the compression baselines (the first / second items represent random and magnitude pruning). Based on the results shown above, we find our strategies generally outperform the model compression baselines and these results support our conclusion in the draft.

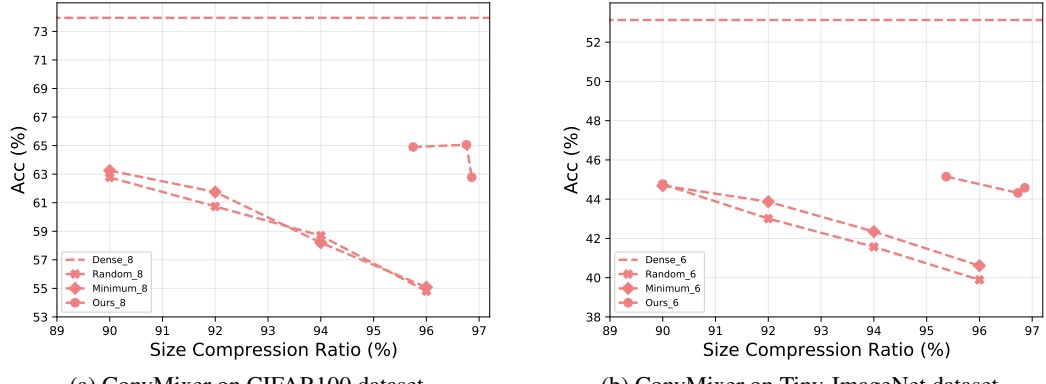

(a) ConvMixer on CIFAR100 dataset.

(b) ConvMixer on Tiny-ImageNet dataset.

Figure 1: Compression performance validation on CIFAR100/Tiny-ImageNet datasets on ConvMixer backbone. Y-axis denotes the test accuracy. X-axis means the network size compression ratio. The straight lines on the top are performance of dense model with regular training. Lines with different symbol shapes denote different settings. Our three points are based on *RP 1e-1*, *RP 1e-2*, and *RP 1e-3*, respectively. This figure shows that our proposed paradigm achieves admirable compression performance compared with baselines. We can still maintain the test accuracy in very high compression ratios.

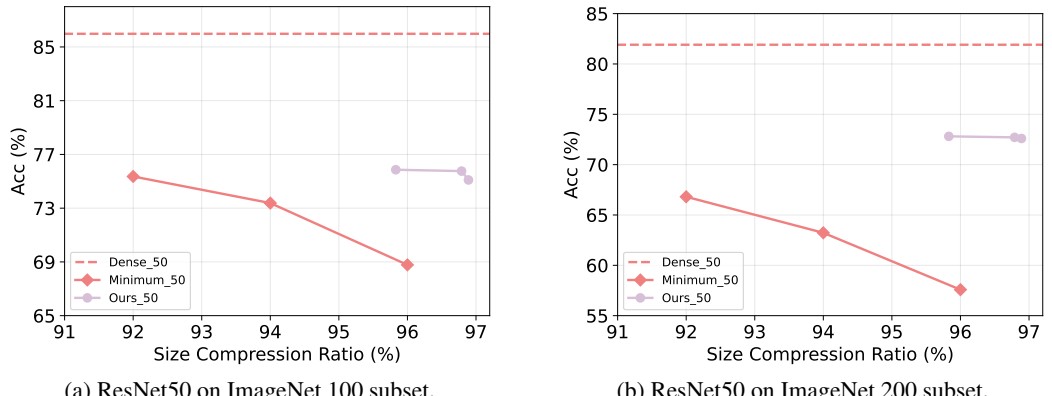

(a) ResNet50 on ImageNet 100 subset.

(b) ResNet50 on ImageNet 200 subset.

Figure 2: Compression performance validation on ImageNet 100/200 subsets on ResNet50 backbone. Y-axis denotes the test accuracy. X-axis means the network size compression ratio. The straight lines on the top are performance of dense model with regular training. Lines with different symbol shapes denote different settings. Our three points are based on *RP 1e-1*, *RP 1e-2*, and *RP 1e-3*, respectively. This figure shows that our proposed compression strategy achieves promising performance on challenging ImageNet dataset compared with baselines.

Table 1: Repeated experimental results for ConvMixer 6-block in subfigure (a) of Fig. 3.

| Dim | Mask | One-layer | MP | RP_1e-1 | RP_1e-2 | RP_1e-3 | RP_1e-4 | RP_1e-5 |
|---|---|---|---|---|---|---|---|---|
| 256 | 89.31 (0.16) | 88.80 (0.11) | 88.97 (0.08) | 88.70 (0.21) | 88.89 (0.13) | 88.52 (0.15) | 85.76 (0.34) | 81.01 (0.38) |
| 512 | 91.90 (0.03) | 91.87 (0.14) | 92.02 (0.02) | 92.07 (0.12) | 92.13 (0.16) | 92.05 (0.03) | 90.55 (0.14) | 87.40 (0.20) |

Table 2: Repeated experimental results for ConvMixer 8-block in subfigure (b) of Fig. 3.

| Dim | Mask | One-layer | MP | RP_1e-1 | RP_1e-2 | RP_1e-3 | RP_1e-4 | RP_1e-5 |
|---|---|---|---|---|---|---|---|---|
| 256 | 90.42 (0.09) | 90.47 (0.04) | 90.65 (0.11) | 90.63 (0.14) | 90.59 (0.06) | 90.06 (0.22) | 87.64 (0.19) | 82.34 (0.26) |
| 512 | 92.69 (0.11) | 92.71 (0.06) | 93.21 (0.05) | 92.90 (0.07) | 92.88 (0.15) | 92.90 (0.07) | 91.71 (0.20) | 87.40 (0.22) |

Table 3: Repeated experimental results for ViT 6-block in subfigure (c) of Fig. 3.

| Dim | Mask | One-layer | MP | RP_1e-1 | RP_1e-2 | RP_1e-3 | RP_1e-4 | RP_1e-5 |
|---|---|---|---|---|---|---|---|---|
| 256 | 76.35 (0.15) | 76.21 (0.14) | 76.70 (0.10) | 77.01 (0.17) | 76.76 (0.21) | 76.80 (0.20) | 64.84 (0.21) | 65.76 (0.23) |
| 512 | 80.73 (0.21) | 81.56 (0.25) | 81.50 (0.11) | 81.87 (0.06) | 81.25 (0.13) | 80.98 (0.16) | 79.17 (0.25) | 79.00 (0.14) |

Table 4: Repeated experimental results for ViT 8-block in subfigure (d) of Fig. 3.

| Dim | Mask | One-layer | MP | RP_1e-1 | RP_1e-2 | RP_1e-3 | RP_1e-4 | RP_1e-5 |
|---|---|---|---|---|---|---|---|---|
| 256 | 79.21 (0.09) | 79.54 (0.16) | 79.23 (0.22) | 79.30 (0.08) | 79.62 (0.15) | 79.28 (0.14) | 73.79 (0.26) | 71.71 (0.28) |
| 512 | 83.44 (0.17) | 83.50 (0.26) | 83.66 (0.12) | 83.67 (0.13) | 83.25 (0.20) | 83.34 (0.19) | 76.73 (0.31) | 78.84 (0.29) |

Table 5: Repeated experimental results for ResNet56/32 on CIFAR10 in subfigure (a) of Fig. 4.

| Network | pr0.9 | pr0.92 | pr0.94 | pr0.96 | Ours-MP | Ours-RP_1e-1 | Ours-RP_1e-2 |
|---|---|---|---|---|---|---|---|
| ResNet56 | 86.35 (0.22) / 86.74 (0.28) | 85.55 (0.14) / 86.01 (0.17) | 85.01 (0.19) / 84.62 (0.27) | 81.93 (0.16) / 82.04 (0.18) | 88.13 (0.19) | 88.36 (0.24) | 87.97 (0.21) |
| ResNet32 | 84.21 (0.13) / 84.22 (0.05) | 83.29 (0.11) / 83.60 (0.20) | 82.08 (0.16) / 81.56 (0.28) | 79.36 (0.13) / 79.12 (0.15) | 86.33 (0.14) | 86.58 (0.09) | 86.39 (0.18) |

Table 6: Repeated experimental results for ResNet56/32 on CIFAR100 in subfigure (b) of Fig. 4.

| Network | pr0.9 | pr0.92 | pr0.94 | pr0.96 | Ours-MP | Ours-RP_1e-1 | Ours-RP_1e-2 |
|---|---|---|---|---|---|---|---|
| ResNet56 | 55.21 (0.36) / 54.01 (0.38) | 51.96 (0.33) / 52.54 (0.16) | 49.72 (0.18) / 49.70 (0.25) | 44.00 (0.13) / 44.18 (0.26) | 56.39 (0.29) | 56.58 (0.21) | 55.84 (0.27) |
| ResNet32 | 49.21 (0.29) / 49.35 (0.14) | 47.36 (0.11) / 47.38 (0.07) | 43.70 (0.41) / 44.60 (0.29) | 39.78 (0.23) / 39.54 (0.37) | 51.76 (0.25) | 50.78 (0.30) | 51.94 (0.19) |