# OpenReview forum: "Parameter-Efficient Masking Networks"
_NeurIPS.cc/2022/Conference — NeurIPS 2022 Accept_

### Official Review · Reviewer_9nMH · 2022-07-10

**Rating:** 6
**Confidence:** 4
**Soundness:** 3 good
**Presentation:** 2 fair
**Contribution:** 3 good

**Summary:**

This paper proposes a new way of representing a neural network in a compressed way coined "One Layer is All You need". The idea is to keep a single fixed and randomly initialized weight vector as prototype for each layer of the network, whereas each layer is saved as a learned mask determining which weights of the prototype are used. Since saving bit masks is more memory efficient than floating point, the network can be efficiently stored. Experiments with ResNet32, ResNet56, ConvMixer and ViT on CIFAR10 and CIFAR100 show that this method achieves improved results in terms of accuracy compared to sparse network training baselines while maintaining larger compression ratios.

**Questions:**

- {Q1} The paper mentions a predefined sparse selection ratio (line 129) which is not further mentioned in the experiments. What value did you set K to for your experiments?
- {Q2} Given the paper, it is not clear to me how the compression ratio is computed for MP/RP in Figures 4/5. I understand that Eq. 10 is used for previous work, however, the proposed method seems to decouple p in C_w (which depends on the fixed vector size) and C_m (which depends on K as mentioned in Q1). Could you please show how you calculated the compression sizes with an example?

I'm willing to raise my score if my questions and concerns ({W2}) are addressed.

**Limitations:**

The authors discussed limitations in the supplementary material. The fact that this method cannot be used to compress already pretrained models but requires training from scratch is an important limitation and should be mentioned in the main paper.

**Strengths And Weaknesses:**

Strengths:
- {S1} The problem of storing neural networks in an efficient manner is significant and the proposed idea improves in this direction.
- {S2} The trade-off between network compression and accuracy is improved in comparison to sparse network training baselines.
- {S3} The writing is well-structured and easy to follow.

Weaknesses:
- {W1} Experiments only performed on low-resolution datasets (CIFAR10, CIFAR100, TinyImagenet).
- {W2} It is not clear if experimental settings are repeated with different seeds. The checklist refers to the supplementary material, but I cannot find any results for multiple seeds there either. I believe all experiments should be conducted for multiple seeds.
- {W3} No code is included for reproducibility.
- {W4} The writing should be improved in terms of typos and grammar (see below for some instances).

Typos:
- The sentence in lines 20-21 seems incomplete.
- Lines 126/150: rewrited -> rewritten
- Line 259: compreesion -> compression
- Line 263: The sentence is confusing, because you train two networks with different strategies and not one network with both.
- Line 307: foundamental -> fundamental

--------------------------------------------------------------------------

{W2} is addressed by the authors during the discussion. Furthermore, they ensured they will resolve {W3} and {W4}. I updated my score respectively.

---

> ### Author Response · Authors · 2022-08-02
> **The Response to the Reviewer 9nMH (Part1)**
>
> # Response to the Reviewer 9nMH:
> We appreciate the reviewer's recognition of our study and detailed comments to help us make further improvement for our submission. We summarize the reviewer’s comments and respond to them as below.
>
> ## Experiments on high-resolution dataset:
> We agree with the reviewer that adding more challenging datasets (e.g, high-resolution image datasets) is very helpful to improve our work.
> However, since our current work aims to explore a scientific question and propose a novel prototol for model compression, it is relatively hard to conduct more fine-grained evaluations from a fair and systematical aspect.
> Please refer to our response to the Reviewer jejx for more discussion details about this point and we leave such kinds of exploration in our future work.
>
> ## Repeated experiments:
> We thank the reviewer pointing it out. We supplement the repeated experimental results as below with necessary analysis. Due to the rebuttal time limitation, we firstly supplement the main results from Figure.3 and Figure.4 to address the reviewer's concern. Please note due to the 9-page limitation of draft revision, we did not change the original draft and make response here. We repeated each experiment three times and report mean and std.
>
>
> **Section.4.2:**
>
> **Figure.3**
>
> **subfigure (a):**
> ConvMixer: 6-block
>
> |Dim| Mask|One-layer|MP|RP_1e-1|RP_1e-2|RP_1e-3|RP_1e-4|RP_1e-5|
> | :----: | :-----: | ------------ | :-----: | :-----: |:-----: |:-----: |:-----: |:-----: |
> | 256| 89.31 (0.16) | 88.80 (0.11) | 88.97 (0.08) | 88.70 (0.21) | 88.89 (0.13) | 88.52 (0.15)| 85.76 (0.34)| 81.01 (0.38)|
> | 512| 91.90 (0.03) | 91.87 (0.14) | 92.02 (0.02) | 92.07 (0.12)| 92.13 (0.16) | 92.05 (0.03) | 90.55 (0.14)| 87.40 (0.20)|
>
> **subfigure (b):**
> ConvMixer: 8-block
>
> |Dim| Mask|One-layer|MP|RP_1e-1|RP_1e-2|RP_1e-3|RP_1e-4|RP_1e-5|
> | :----: | :-----: | ------------ | :-----: | :-----: |:-----: |:-----: |:-----: |:-----: |
> | 256| 90.42 (0.09) | 90.47 (0.04) | 90.65 (0.11) | 90.63 (0.14) | 90.59 (0.06) | 90.06 (0.22) | 87.64 (0.19) | 82.34 (0.26)
> | 512| 92.69 (0.11) | 92.71 (0.06) | 93.21 (0.05) | 92.90 (0.07) | 92.88 (0.15) | 92.90 (0.07) | 91.71 (0.20) | 87.40 (0.22)
>
> **subfigure (c):**
> ViT: 6-block
>
> |Dim| Mask|One-layer|MP|RP_1e-1|RP_1e-2|RP_1e-3|RP_1e-4|RP_1e-5|
> | :----: | :-----: | ------------ | :-----: | :-----: |:-----: |:-----: |:-----: |:-----: |
> | 256| 76.35 (0.15) | 76 21 (0.14) | 76.70 (0.10) | 77.01 (0.17) | 76.76 (0.21) | 76.80 (0.20) | 64.84 (0.21) | 65.76 (0.23)
> | 512| 80.73 (0.21) | 81.56 (0.25) | 81.50 (0.11) | 81.87 (0.06) | 81.25 (0.13) | 80.98 (0.16) | 79.17 (0.25) | 79.00 (0.14)
>
> **subfigure (d):**
> ViT: 8-block
>
> |Dim| Mask|One-layer|MP|RP_1e-1|RP_1e-2|RP_1e-3|RP_1e-4|RP_1e-5|
> | :----: | :-----: | ------------ | :-----: | :-----: |:-----: |:-----: |:-----: |:-----: |
> | 256| 79.21 (0.09) | 79.54 (0.16) | 79.23 (0.22) | 79.30 (0.08) | 79.62 (0.15) | 79.28 (0.14) | 73.79 (0.26) | 71.71 (0.28)
> | 512| 83.44 (0.17) | 83.50 (0.26) | 83.66 (0.12) | 83.67 (0.13) | 83.25 (0.20) | 83.34 (0.19) | 76.73 (0.31) | 78.84 (0.29)
>
> Based on the results shown above, we make several conclusions: 1) Our experimental performance are generally stable across different datasets and different settings. The supplemented results follow the accuracy patterns and support the conclusions provided in our draft; 2) ConvMixer is more stable than ViT backbone across different settings; 3) Overall, along with the decreasing number of unique values in the network (from the left column to the right column of tables), the performance variations increase correspondingly. The limited unique weight values decreases the stability of the network.

---

> ### Author Response · Authors · 2022-08-02
> **The Response to the Reviewer 9nMH (Part2)**
>
> **Section.4.3:**
>
> **Figure.4**
>
> **subfigure (a):**
> ResNet56/ResNet32: CIFAR10
>
> |Network| pr0.9 |  pr0.92 | pr0.94 | pr0.96 | Ours-MP | Ours-RP_1e-1| Ours-RP_1e-2 |
> | :----: | :-----: | ------------ | :-----: | :-----: |:-----: |:-----: |:-----: |
> | ResNet56| 86.35 (0.22) // 86.74 (0.28)| 85.55 (0.14) // 86.01 (0.17) | 85.01 (0.19) // 84.62 (0.27) | 81.93 (0.16) // 82.04 (0.18) | 88.13 (0.19)| 88.36 (0.24)| 87.97 (0.21)|
> | ResNet32| 84.21 (0.13) // 84.22 (0.05)| 83.29 (0.11) // 83.60 (0.20) | 82.08 (0.16) // 81.56 (0.28) | 79.36 (0.13) // 79.12 (0.15) |  86.33 (0.14) |  86.58 (0.09) |  86.39 (0.18) |
>
> **subfigure (b):**
> ResNet56/ResNet32: CIFAR100
>
> |Network| pr0.9 |  pr0.92 | pr0.94 | pr0.96 | Ours-MP | Ours-RP_1e-1| Ours-RP_1e-2 |
> | :----: | :-----: | ------------ | :-----: | :-----: |:-----: |:-----: |:-----: |
> | ResNet56| 55.21 (0.36) // 54.01 (0.38) | 51.96 (0.33) // 52.54 (0.16) | 49.72 (0.18) // 49.70 (0.25) | 44.00 (0.13) // 44.18 (0.26)| 56.39 (0.29) | 56.58(0.21) | 55.84 (0.27) |
> | ResNet32| 49.21 (0.29) // 49.35 (0.14) | 47.36 (0.11) // 47.38 (0.07) | 43.70 (0.41) // 44.60 (0.29) | 39.78 (0.23) // 39.54 (0.37)| 51.76 (0.25) | 50.78 (0.30) | 51.94 (0.19) |
>
> In the tables above, the first four columns show the compression baselines (the first // second items represent random and magnitude pruning).
> Based on the results shown above, we find our strategies generally outperform the model compression baselines and these results support our conclusion in the draft.
>
> ## Code release:
> Our code will be rearranged and released for reproducibility.
>
> ## Writing:
> We appreciate the reviewer's careful reading and pointing out our typos. We will polish our draft to fix these typos and unclear descriptions for a better version.
>
> ## Sparse selection ratio K:
> We follow the Popup method to set ratio K as 0.5, which is the best ratio to conduct the sparse selection. All the experiments are based on ratio K=0.5. Thanks for pointing out our missed details and we will clarify this point in our final version.
>
> ## Computational details about compression ratio of MP/RP:
> We thank the reviewer's careful reading.
> We use a demo numerical example to illustrate the calculations of the compression ratio for our strategies.
> We take our methods on ConvMixer with 6-block/256-dim in Figure.5 (in purple color) as an example: (1) If we initialize the model randomly (the regular training setting), we assume all initialized parameters are unique. In this case, this number is 432460; (2) We start from our "MP" strategy (using the largest layer as prototype). In this case, the largest layer contains 65536 parameters. Please note our "RP" strategy is termed based on its prototype size compared with "MP" size. The strategies used in this case are "RP 1e-1", "RP 1e-2", and " RP 1e-3" (mentioned in caption of Figure.5). Therefore, their prototype sizes (number of unique values) are $0.1 \times 65536$, $0.01 \times 65536$, and $0.001 \times 65536$, respectively. They are around 6554, 655, and 66; (3) Our compression ratio is compared with the full model size (100%), which requires to restore all 432460 float values. Compared with it, our strategies firstly require restore their prototype as float values (6554, 655, and 66). Correspondingly, they require 6554/432460, 655/432460, and 66/432460 partition of the full size (100%) model. They are 1.5%, 0.15%, and 0.015%. Please note since we sequentially make padding operations to use prototype to fill up the whole network, thus, no additional storage cost needed here. After obtaining the prototype, we need to restore a bunch of masks. Because the prototype constructs the whole model with the same structure as the original one, our masks have exactly the same size accordingly, which have in 432460 numbers in total. We can efficiently restore the binary mask using 1/32 storage cost compared with float values. Thus, this part costs 1/32 partition of the full size (100%). It is 3.1%; (4) We combine these two parts of cost (prototype and masks) as 1.5%+3.1%, 0.15%+3.1%, and 0.015%+3.1%. They are 4.6%, 3.25%, and 3.115%. Converting them to compression ratio, 1-4.6%, 1-3.25%, and 1-3.115%, they are 95.4%, 96.75%, 96.885%, respectively. Correspondingly, the x-axix values of three purple points on the right top corner of Figure.5 represent these three ratios. We take this case as an example and other cases follow the same way to calculate the compression ratio.
>
>
> ## Limitations:
> We will supplement this limitation in our main draft to provide a better understanding of our work.

---

> ### Author Response · Authors · 2022-08-07
> **The Response to the Reviewer 9nMH (Part3)**
>
> ## Supplementary repeated experiments
> We further supplement the repeated empirical results for the rest of our experiments.
>
> **Figure.5**
> ConvMixer 256-dim: CIFAR10
>
> |Depth| pr0.9 |  pr0.92 | pr0.94 | pr0.96 | Ours-RP_1e-1 | Ours-RP_1e-2| Ours-RP_1e-3 |
> | :----: | :-----: | ------------ | :-----: | :-----: |:-----: |:-----: |:-----: |
> | 6 | 84.34 (0.19) // 84.45 (0.16) | 83.08 (0.26) // 83.31 (0.25) | 81.17 (0.30) // 80.99 (0.23) | 77.10 (0.35) // 77.34 (0.28)| 88.70 (0.21) | 88.89 (0.13) | 88.52 (0.15) |
> | 8 | 86.93 (0.14) // 86.88 (0.28) | 85.30 (0.32) //  84.98 (0.11) | 83.37 (0.35) // 83.72 (0.26) | 79.88 (0.27) // 79.79 (0.29)| 90.63 (0.14) | 90.59 (0.06) | 90.06 (0.22) |
>
> Please note that we added compression baselines results in the first four columns (first/second terms are random/magnitude pruning). And we directly refer to the results provided above (repeated results for subfigure (a) and subfigure (b) of Figure.3). The newly added results also support our conclusions.
>
> **Supplementary Figure.1**
>
> **Subfigure (a):**
> ConvMixer 256-dim: CIFAR100
>
> |Depth| pr0.9 |  pr0.92 | pr0.94 | pr0.96 | Ours-RP_1e-1 | Ours-RP_1e-2| Ours-RP_1e-3 |
> | :----: | :-----: | ------------ | :-----: | :-----: |:-----: |:-----: |:-----: |
> | 8 | 63.27 (0.16) // 62.95 (0.24) | 61.80 (0.19) //  61.02 (0.34) | 58.10 (0.28) // 58.72 (0.30) | 55.35 (0.29) // 54.66 (0.23)| 64.78 (0.24) | 64.96 (0.18) | 62.83 (0.27) |
>
> **Subfigure (b):**
> ConvMixer 256-dim: Tiny-ImageNet
>
> | Depth | pr0.9 |  pr0.92 | pr0.94 | pr0.96 | Ours-RP_1e-1 | Ours-RP_1e-2| Ours-RP_1e-3 |
> | :----: | :-----: | ------------ | :-----: | :-----: |:-----: |:-----: |:-----: |
> | 6 | 44.72 (0.23) // 44.58 (0.16) | 42.88 (0.36) // 43.98 (0.21) | 41.73 (0.29) // 42.24 (0.17) | 40.10 (0.22) // 40.78 (0.34) | 45.31 (0.27) | 44.29 (0.21) | 44.56 (0.33) |
>
> We also supplement the repeated results for experiments in the supplementary material. They are based on ConvMixer for CIFAR100 and Tiny-ImageNet, respectively. The first/second terms are random/magnitude pruning results in the first four columns. The newly added experimental results also support our conclusion in the draft.

---

> ### Author Response · Authors · 2022-08-07
> **Sincerely Expecting Discussions with the Reviewer 9nMH**
>
> Dear Reviewer 9nMH,
>
> We appreciate the reviewer's recognition of our work and providing detailed comments for us to make further improvement! For the reviewer's concerns and comments, we have responded accordingly with supplemented experimental results and necessary discussions or clarifications. Given the NeurIPS final discussion deadline (08/09) is approaching, we really hope to have a further discussion with the reviewer 9nMH to see if our responses solve the reviewer's concerns.
>
> Thank you very much for your time!
>
> Best wishes,
>
> Authors of Paper1207

---

> ### Comment · Reviewer_9nMH · 2022-08-08
> **Response**
>
> I thank the reviewers for their thorough answers. My main concern {W2} is resolved and I'll update my score. I still think that adding an experiment on high resolution data {W1} (e.g. ImageNet) would be beneficial for the camera ready version. I do not see how such an experiment would be unfair in terms of comparisons to your own baselines. Lastly, I believe that exploring a scientific question should also include scalability.

---

> > ### Author Response · Authors · 2022-08-09
> > **The Response of the Reviewer 9nMH**
> >
> > Dear Reviewer 9nMH,
> >
> > We appreciate the reviewer's recognition and support of our work and rebuttal. We also thank the reviewer's suggestions to help us further improve our work for both scientific exploration and compression evaluation. We will prepare them accordingly for our draft to deliver a better final version.
> >
> > Thank you very much for your time!
> >
> > Best wishes,
> >
> > Authors of Paper1207

---

### Official Review · Reviewer_jejx · 2022-07-11

**Rating:** 7
**Confidence:** 4
**Soundness:** 3 good
**Presentation:** 3 good
**Contribution:** 3 good

**Summary:**

This paper proposed a new paradigm for neural network compression. The authors randomly initialize a set of weights. The actual parameters of each layer are represented as the initialized weights with binary masks. The weights are shared by multiple layers, while the masks are different for each layer. The weights are fixed, while the masks are learnable. In this way, the total bytes are significantly reduced. Experiments show that the proposed method achieves better compression than baselines.

**Questions:**

None (refer to weaknesses)

**Limitations:**

Authors discussed the limitations and potential negative societal impact of their work in supplementary material.

**Strengths And Weaknesses:**

Strengths:

1. This paper is well organized, and the core method is clearly represented.
2. This paper represents each layer as shared weights with different masks. The idea for model compression is interesting and novel.
3. Experiments show that the proposed method achieves good compression for image classification models.

Weaknesses:

1. The title of this paper is unsuitable and the authors should change it. First, people will not associate the title with model compression. Second, the word "one layer" in this paper is misleading. Although some parts are shared cross all layers, there are differences between layers. Thus, we can't say them "one layer". In my opinion, masks are also parameters of the model.
2. It is better to compare the compression performance with stronger baselines or bigger datasets such as ImageNet.
3. In general, the proposed method achieves compression by sharing some parts of parameters (while adjusting the others). Several previous works have explored this direction, such as [1] and [2]. The authors should discuss them in related works.

[1] Residual connections encourage iterative inference. International Conference on Learning Representations, 2017

[2] Recurrent convolutions: A model compression point of view. NIPS Workshops: Compact Deep Neural Network Representation with Industrial Applications, 2018

---

> ### Author Response · Authors · 2022-08-02
> **The Response to the Reviewer jejx (Part1)**
>
> # Response to the Reviewer jejx:
> We appreciate the reviewer‘s acknowledge of our novelty and detailed comments for us to make further improvement. We summarize the reviewer's comments and make responses as below.
>
> ## Title revision:
> We thank the reviewer pointing out the problem of our current title with detailed suggestions. This point is also mentioned by other reviewers. As the reviewer mentioned, the "one layer" causes some confusions and may mislead the understanding of our work. We revised our title as **Iterative Mask Learning on Limited Random Weights** to convey the key factors of our work. Please refer to our responses to the reviewer vyrt and the reviewer PU7f for more discussion details about our title revision.
>
> ## More comparisons:
> We thank the reviewer's constructive comments for our experiments.
> For the stronger baselines, since our work aims to proposes a new protocol for model compression by representing a sparse model with a small-scale random vector and different masks, it is different from typical model compression framework, which requires to restore the float values and position of sparse networks.
> In addition, previous model compression works follow different settings.
> For example, the classic pruning methods are based on pretrained model and recent pruning at initialization methods are based on random initialized network.
> Another instance is that some compression methods globally pick the unimportant weights to remove based on given criteria, which causes different layer-wise sparsity patterns.
> But some others pre-define the sparse ratio for each layer (e.g., consistent layer-wise ratio).
> Therefore, it is relatively hard to make a fair and systematical comparsion with more recent advanced compression approches based on our current exploration.
> In our current version, we construct two baselines with random pruning and magnitude-based pruning which is simple but commonly effective approach for comparison.
> It demonstrates the effectiveness of our proposed protocol.
> We expect our work can inspire more detailed explorations in this direction.
> We leave more fine-grained explorations based on different settings mentioned above in our future works.
> For other dataset comparisons, we have included the Tiny-imagenet dataset in our supplementary material for large-scale dataset validation. Similarly, we leave the explorations of more challenging datasets with different tasks in our future works.

---

> ### Author Response · Authors · 2022-08-02
> **The Response to the Reviewer jejx (Part2)**
>
> ## Related works:
> We appreciate the reviewer providing more related works of our study, which are missed by our current version.
> Both of them are relevant and valuable with insightful ideas. Due to the 9-page limitation of the revision draft, we discuss these papers in the rebuttal here and we will integrate these dicussions into our final version for a better literature review.
> Concrete discussions are shown as below:
>
>
>
> - **Residual connections encourage iterative inference. International Conference on Learning Representations, 2017:**
> This paper aims to understand the effective computational machenism of ResNet architecture. It formalize a perspective to study the iterative feature refinement achieved by the residual archtecture and understand the optimization behavior.
> It shows the residual block encourages the feature to move along the negative gradient of loss. It also empirically shows that the shallow layers in ResNet focus more on representation learning and deeper layers focus on iterative refinement of the learned features. Leveraging on the iterative refinement perspective, this paper explores the residual layer weights sharing strategy. It finds training residual blocks with weights sharing leads to overfitting and proposes a batch normalization based approach to handle this issue. Different from this work, our study explores the representative capacity of given random weights by iteratively learning different masks on them. In this way, the fixed random weights can deliver diverse feature mappings. Naturally, a new model compression paradigm is proposed along with our exploration of the random weights representative capacity.
>
> - **Recurrent convolutions: A model compression point of view. NIPS Workshops: Compact Deep Neural Network Representation with Industrial Applications, 2018:**
> This work focuses on efficiently using the recurrent convolution (RC) kernels to conduct model compression. Specifically, it uses the same RC kernel and unrolls it multiple times to reduce the layer-wise redundancy in the network. To tackle the performance drop caused by the RC kernel sharing, this paper wisely designs a simple yet effective strategy based on a variant of batch normalization. This variant enables the BN layers of RC kernels can be learned independently. In this way, the performance drop caused by the weights sharing can be recovered. Further, unrolling RC kernels for networks can be employed as a practical strategy for model compression usage. Different from this work, our study explores the capacity random fixed weight, which covers more general network cases (e.g., CNN, Transformer). In addition, leveraged on the proposed new protocol for compression, our compression strategy studies more fine-grained weight sharing with limited unique weight values. The different masks can be conducted on the given fixed weights to output diverse feature mappings. In this way, the model storage is more efficient compared with typical compression approaches and enables us to make model compression under more extreme cases.

---

> ### Comment · Reviewer_jejx · 2022-08-08
> **Update my score to 7.**
>
> Most of my concerns are addressed by the authors' response.
> Although they don't compare with stronger baselines in the current version, the reasons claimed by the authors are convincing to me.
>
> What is the representative capacity of random weights? The authors make a meaningful exploration to this fundamental question. When I first review this paper, I focus on the pruning but somewhat ignore the scientific question behind it.
>
> So I improve my score from 6 to 7.

---

> > ### Author Response · Authors · 2022-08-09
> > **The Response to the Reviewer jejx**
> >
> > Dear Reviewer jejx,
> >
> > We appreciate the reviewer's recognition of our rebuttal, careful reading of our draft, and further support of our work. We will reorganize our draft and emphasize our key points and contributions to deliver a better final version.
> >
> > Thank you very much for your time!
> >
> > Best wishes,
> >
> > Authors of Paper1207

---

### Official Review · Reviewer_PU7f · 2022-07-11

**Rating:** 8
**Confidence:** 4
**Soundness:** 4 excellent
**Presentation:** 3 good
**Contribution:** 3 good

**Summary:**

This paper aims to handle the difficulty of restoring/transmitting models caused by the increasing model size for recent large-scale neural networks. Inspired by recent works (e.g., LTH, Popup) on random networks, the paper starts by answering a scientific question: what is the potential of a random network? Specifically, the authors propose a series of strategies to study the random network with different masks to map different features. Through the exploration for the answer, a new model compression paradigm is proposed by only restoring one-layer random weights and a bunch of masks to represent a model. Experiments were conducted based on using different CNN/transformer architectures. Extensive results validate the rationality of the motivation and show the feasibility of the new compression paradigm.

**Questions:**

See the above weaknesses.

**Limitations:**

The authors have addressed the limitations and potential negative societal impact.

**Strengths And Weaknesses:**

Strengths:
1) This work tries to reduce the model storage size, which is a clear and practical motivation. Compared with typical model size compression methods that remove partial parameters, it is a novel way to represent a model by using different masks on fixed random weights.
2) This work is driven by studying the random weight capacity, which is an interesting yet under-explored studying point. It is novel to use “one-layer” weights with different masks to learn a model.
3) Experiment is extensive using different model architectures. Firstly, it answers the question about random weight potential using a series of proposed strategies to construct a network using random weights. Secondly, it shows the feasibility of a new compression paradigm compared with the typical model compression method.

Weakness:
1) It is encouraged to revise the draft title to a more appropriate one. After reading the draft, I think the current title doesn’t convey the key factor of this paper. Iteratively selecting different masks on a set of fixed random weights for different feature mappings should be the main point, therefore, the usage of “one-layer” in the title is inaccurate. On the other side, “all you need” is a too vague description. It needs to be concretized to eliminate confusion.
2) Some related works are supplemented in the appendix, I suggest moving them into the main draft and providing necessary discussions about them. The discussion should include the difference between the submitted work with these existing works since they look highly related to this work, even if they are in a different setting.
3) Technically, the proposed random vector padding (RP) repeats the given set of random weights in the same order. If randomly shuffling the random set and then doing the padding to construct the model, can it improve the capacity?
4) Minors: (1) In Alg.3, it seems the output is written in the wrong way, which should be the output of MP strategy in Alg.2, but not consistent with RP strategy in Alg.3. (2) Around Eq. 5 and Eq. 6, the explanation of T^p is missing. It should be further clarified and consistent with Alg.1. (3) In Eq.9, the d_l should be the dimension of w_l instead of the number of vector v_pro. Please make it clear to eliminate confusion.

---

> ### Author Response · Authors · 2022-08-02
> **The Response to the Reviewer PU7f**
>
> # Response to the Reviewer PU7f:
> We appreciate the reviewer's recognition of our work and valuable suggestions for us to make further improvement. We summarize the reviewer’s comments and respond to them as below.
>
> ## Title revision:
> We agree with the reviewer that our current title may cause some confusions and needs revision. This point is also mentioned by other reviewers. Accordingly, we revised our title as “Iterative Mask Learning on Limited Random Weights”, which covers the key factors of our work. Please refer to our response to the Reviewer vyrt for more details about the title revision.
>
> ## Related works:
> We agree with the reviewer that the related works mentioned in the supplemenary are necessary to be discussed in the main file. These works are based on different settings for different tasks, however, they also conceive the insight of weight reusing based on several iterative strategies. We will add more discussions and integrate them into our related work section.
>
> ## Shuffling random weights:
> This is a good point to explore. We further added some experiments to show the effectiveness of shuffling random weights. These experiments are based on ConvMixer backbone with 6/8 blocks and 256/512 dimensions. We use two RP strategies as RP_1e-1 and RP_1e-2. The results are shown below.
> We refer to the padding results from our draft.
>
> |RP_1e-1| 6/256|8/256|6/512|8/512|
> | :----: | :-----: |:-----: |:-----: |:-----: |
> | Padding|   89.09 |90.64 | 92.24 | 92.99 |
> | Shuffling| 89.27 | 90.69  | 91.61  | 92.94 |
>
> |RP_1e-2| 6/256|8/256|6/512|8/512|
> | :----: | :-----: |:-----: |:-----: |:-----: |
> | Padding| 88.96 | 90.72 | 92.16 | 93.09 |
> | Shuffling| 89.23 | 90.59 |92.04 | 93.00 |
>
>
> Based on the results shown above, we found there is no consistent and significant changes about performance compared between padding and shuffling strategies. However, the random shuffling will cause additional storage cost as the prototype order has been changed, which is not promising for model compression. We appreciate the reviewer's technically comments for this interesting point. We leave the further exploration of how to leverage more on shuffling the prototype to gain improvement in our future works.
>
>
> ## Minors and typos:
> We appreciate the reviewer's careful reading and pointing our typos. We will further polish our draft to deliver a better version.

---

> > ### Comment · Reviewer_PU7f · 2022-08-07
> > **Response**
> >
> > Thanks for your response. My concerns have been addressed in the response. I think it is a novel work with extensive evaluations.

---

> > > ### Author Response · Authors · 2022-08-07
> > > **The Response to the Reviewer PU7f**
> > >
> > > Dear Reviewer PU7f,
> > >
> > > We appreciate the reviewer's recognition of our rebuttal and the further support of our work.
> > >
> > > Thank you very much for your time!
> > >
> > > Best wishes,
> > >
> > > Authors of Paper1207

---

### Official Review · Reviewer_vyrt · 2022-07-13

**Rating:** 6
**Confidence:** 3
**Soundness:** 3 good
**Presentation:** 3 good
**Contribution:** 2 fair

**Summary:**

The paper studies the use of random weights together with learnable masks. The learnable masks are learned with straight through operator. The authors argue that such training approach for neural network would reduce the model storage requirements and has applications to network compression. The model is validated on cifar 10 and cifar 100 datasets showing that the proposed layers underperforms (approximately) between 1 and 10 accuracy points wrt dense layers depending on the model architecture and number of parameters.

-------
Post rebuttal: Based on authors responses I updated the overall score from 3 to 6 and increased soundness and presentation both by 1.

**Questions:**

See above (strengths and weaknesses).

**Limitations:**

- The paper does not discuss the limitations of the discussed ideas.

- The paper does not mention societal impacts.


**Strengths And Weaknesses:**

Overall, the discussed ideas are interesting – using a random layer with learnable masks to achieve competitive performance. However, the validation and the presentation of the paper require improvements. For details see comments below.

**Title**
- The title might be a bit to generic (not very informative) and a bit misleading (for more complex datasets one would probably need more layer architectures). How about the following title:  On learning masking operators for network random weights.

**Abstract**
- To strengthen the abstract, please add quantification of improvements in terms of space complexity. Also, based on the experimental section the improvements come at the expense of model accuracy, however, this trade-off is not captured in the abstract.

**Introduction**
- Introduction section is in general well written and easy to follow. However, it could benefit from some re-writing, shortening, and refocusing.

- The introduction does not discuss the obtained results making it hard to assess the significance of the proposed approach. It is also unclear what the ML community gains with the results of this study. Please add such discussion to the introduction section.

**Methodology**

- This section would benefit the most from re-writing and restructuring. In the current form it is difficult to follow and do not allow to fully appreciate the presented ideas.

- Figure 2 should be better discussed and better formatted. Please extend the caption to clarify the figure. In general, all figures in the paper should be self-explanatory making it possible to understand the figures just by reading the captions. In its current form it is impossible to understand the drawings.

- Based on the description the process of updating the prototype weights into target weights is unclear. Could the authors clarify how different networks are updated?

- Eq 4 and Eq 7 differ only in the dimensionality of w. Why changing the dimensionality leads to new paradigm of random weights padding? Moreover, please boldface vectors in Eq 7 to differentiate them further from scalars in Eq. 4.

- The section lacks motivations behind different choices.

**Results**

- The introduced layer is not compared to previously published models. Would it be possible to compare the model to Supermaks and Popup? Adding comparisons to previous art would make the validation stronger.

- Cifar datasets are small scale. Would the observations generalize to larger scale datasets? Adding another dataset would make the observations more compelling.

- The results are missing stds, making it hard to assess the significance of the results.

- In general, the proposed layers are underperforming w.r.t dense layers. That would be expected. However, the results section lacks discussion and positioning of the reported results, e.g., Why the reported results are interesting? What do we learn as a community from the results? What is the impact of the reported results? Adding more in-depth discussion would make the paper stronger.

---

> ### Author Response · Authors · 2022-08-02
> **The Response to the Reviewer vyrt (Part1)**
>
> # Response to the Reviewer vyrt:
>
> We appreciate the reviewer’s detailed suggestions to help us improve our work. We summarize the reviewer’s comments and make responses as below. Please note that we make careful revisions for several sections based on the suggestions of the reviewer. However, the draft revision has the 9-page limitation which makes it hard to show the revision changes. Therefore, we make response here and leave the draft unchanged.
>
>
> ## Inappropriate title:
>
> Our current title “One Layer is All You Need” was expected to deliver that one layer (or a vector in an even smaller granularity) with fixed random parameter values can represent diverse feature mappings.
>
> We admit this title cannot provide clear and enough information about our research work and needs to be revised for clarity. This point is also suggested by other reviewers. We thank the reviewer’s suggestion, “On learning masking operators for network random weights”, for our title revision. Based on this suggestion, we revise our title as *“Iterative Mask Learning on Limited Random Weights''*. In this way, the title contains necessary factors of our work: 1) We are given a set of fixed random weights, which is small-scale compared with the whole network structure; 2) We iteratively learn different masks on the network augmented by the given random weights for different feature mappings. This version describes our scientific exploration process and the “limited random weights” also indicates the model compression function of our work.
>
>
> ## Abstract revision:
>
> We appreciate the review’s suggestions. It is constructive for a more comprehensive abstract to describe the whole picture of our work. Accordingly, we revised our abstract below where the main revisions are emphasized by boldtype (please note that we also did some trivial polishing but without boldtype).
>
> *Revised abstract:
> A deeper network architecture generally handles more complicated non-linearity and performs more competitively. Recently, advanced network designs often contain a large number of repetitive structures (e.g., Transformer). They empower the network capacity to a new level but also increase the model size inevitably, which is unfriendly to either model restoring or transferring. In this study, **we are inspired by previous works (e.g., Lottery Ticket [7] and Popup[20]) to study the random network capacity. Following this point,** we are the first to investigate the representative potential of fixed random weights with limited unique values by iteratively learning different masks, leading to a new paradigm for model compression to diminish the model size. Concretely, we utilize one random initialized layer, accompanied with different masks, to convey different feature mappings and represent repetitive modules in a deep network. As a result, the model can be expressed as one-layer with a bunch of masks, which significantly reduces the model storage cost. Naturally, we enhance our strategy by learning masks for a model filled by padding a given random weights sequence. In this way, our method can further lower the space complexity, especially for models without many repetitive architectures. **We scientifically explore the potential of random weights by a series of experimental validations and test our proposed compression paradigm based on different network architectures. Compared with typical compression baselines satisfying more accuracy for compression, our method generally achieves better compression-accuracy trade-off based on different settings such as around 10%/6% improvement on CIFAR10 with 96% compression ratio using ConvMixer and ResNet backbones.***

---

> ### Author Response · Authors · 2022-08-02
> **The Response to the Reviewer vyrt (Part2)**
>
> ## Introduction revision:
> We agree with the reviewer that a more detailed introduction delivers a better understanding of our work. We briefly emphasize the key points of the introduction revision below and we will integrate them with the current content of the draft into our final version.
> Summarization of introduction discussion:
> Our work mainly contains two parts. The first part explores a scientific question about the representative potential of random weights with limited unique values. There is a series of previous works such as LTH, Supermasks, and Popup. Based on a given and fixed random network, they try to study the subnetwork trainability, subnetwork representative capacity, and finding subnetwork with better representative capacity, respectively. Inspired but different from them, we further explore the maximum representative potential of random weights with limited unique values. Answering this question helps us understand the operation mechanism of a neural network: do neural networks need high-level diversified numerical values to represent semantic patterns? or this capacity can also be achieved by a diversified topological combination (realized by subnetwork mask) based on weights with limited unique values. Along with answering this question, the second part proposes a novel model compression paradigm. Different from the typical model compression protocol which requires to restore the sparse positions with their float values, our paradigm represents a sparse network by a small amount of random weights with different binary masks. Experiments demonstrate the promising compression results of our method, which benefits efficient model storage and transferring. Overall, our work inspires us to further understand the network operation mechanism and proposes a novel model compression paradigm.
>
> ## Methodology:
> - **Writing logic:**
> We briefly clarify our writing logic of the methodology section and we will reorganize it in our final version. Our proposed technical approaches have three versions: one-layer, max-layer padding (MP), and random weight padding (RP). The one-layer strategy is naturally inspired by the fact that recent proposed deep frameworks always contain several repetitive modules sharing the same architecture such as transformer models. It shares the weights among all modules with the same structure. However, some parts of the network use unique structure (e.g., final projection layer in the transformer) and different parts of the network may use different structures (e.g., the multi-head attention block and feed-forward block in the transformer). Therefore, we naturally use the layer with the most number of parameters as a prototype (MP strategy) to fill up the whole network. In this way, the number of unique values in the model is limited by the size of the largest layer. Furthermore, we only initialize a random vector as a prototype to fill up the whole network (RP strategy), which further reduces the unique values in the model. In summary, our writing follows the logic of our exploration, which reduces the unique values in a model to answer the scientific question of random weight representative capacity and explore the novel compression paradigm simultaneously.

---

> ### Author Response · Authors · 2022-08-02
> **The Response to the Reviewer vyrt (Part3)**
>
> - **Figure 2:**
> Our current caption misses the explanation of certain elements in Fig. 2 such as “prototype” and the color of the vectors. We will supplement them in our final version to eliminate confusions.
>
> - **Prototype usage:**
> We clarify the usage of our three strategies of using prototype to update network with some demo examples.
>   - **One-layer:**
>     We provided an example in the current draft (L151 - L154).
>     A 5-layer MLP network with dimension (512, 100, 100, 100, 10) contains 4 randomly initialized weights matrices: {$w_1 \in \mathbb{R}^{512 \times 100}$, $w_2 \in \mathbb{R}^{100 \times 100}$, $w_3 \in \mathbb{R}^{100 \times 100}$, $w_4 \in \mathbb{R}^{100 \times 10}$}.
>     In this network, $w_1$, $w_2$, and $w_4$ are prototype as they are matrices with different sizes.
>     $w_3$ is the target of $w_2$ as it has the same size as $w_2$.
>     Thus, the network updated by *one-layer* strategy is {$w_1$, $w_2$, $w_2$, $w_4$}.
>
>   - **Max-layer padding (MP):**
>     We use the same example as above. The ***max-layer*** with the largest size is $w_1$. Thus, we use $w_1$ as prototype to update other layers. We flatten the $w_1$ to obtain a vector with length $512 \times 100$. We use the first $100 \times 100$ elements to replace the original weights in $w_2$ and $w_3$. We use the first $100 \times 10$ elements to replace the weights in $w_4$. Formally, the network updated by ***max-layer*** strategy is {$w_1$, $w_1[:s_2]$, $w_1[:s_3]$, $w_1[:s_4]$}, where $s_2, s_3, s_4$ are the layer size of $w_2$, $w_3$, and $w_4$ (please note the flattening and reshaping operations are omitted for simplicity).
>
>   - **Random vector padding (RP):**
>     We use the same example as above. We set the length of random vector as $r$. Still in $w_1$, we truncate the first $r$ elements and obtains $w_1[:r]$ as the prototype (with corresponding flattening) to fill up the whole network. Let us say $r$ is 100 here. We repeat the prototype 512 and 100 times to fill up $w_1$ and $w_2$, respectively.
>
> - **Eq.4 and Eq.7:**
> We would like to clarify that the rewriting from Eq.4 to Eq.7 is only for the convenient descriptions following Eq.7. We use superscript $d_l$ (in Eq.7) to denote the dimention instead of subscript $l$ (in Eq.4) as the index of layer. The following introductions of max-layer padding (MP) and random-vector padding (RP) can be more convenient in this way. We will clarify this point in our final version to eliminate confusions.
>
> - **Motivations**
> For the motivations of different choices in the methodology section, please refer to our response of **Writing logic** above for more details. We will integrate this discussion into our final version.
>
> ## Results:
> - **Comparisons with Popup and Supermasks:**
> Actually, in our experiments, we follow the Popup method as **sparse selection strategy** to achieve our exploration and have compared with the Popup method. In the part of exploring the random weight representative capacity (Sec.4.2), the Popup results are denoted as **Mask** strategy with circle symbols. For the Supermasks approach, since the Popup aims to achieve better sparse network selection and significantly outperforms the performance of Supermasks, we choose to follow the Popup algorithm which is much more promising and did not compare with Supermasks. We will clarify this point and supplement more discussions about Supermasks in our final version.
>
> - **Other datasets:**
> We have also provided the results on Tiny-imagenet in the supplementary material to support our observation. For more challenging datasets with different tasks, we leave them in our future works. Please refer to our response to the reviewer jejx **(More comparisons)** for more discussions about this point.
>
> - **Std of results:**
> This point is also suggested by the Reviewer 9nMH, please refer to our response to the Reviewer 9nMH for more details.
>
> - **More insight discussions:**
> We appreciate the reviewer's contrustive suggestion to foundamentally improve our work. For the key points of the insight discussions for our work, we have summarized them in our response to the point **Introduction revision** above. Please refer to it for more details and we will integrate these discussions in our final version.
>
> ## Limitations:
> The discussions of both limitations of the idea and the societal impacts have already been included in our supplementary material. Please refer to it for more details.

---

> ### Author Response · Authors · 2022-08-07
> **Sincerely Expecting Discussions with the Reviewer vyrt**
>
> Dear Reviewer vyrt,
>
> We appreciate the Reviewer vyrt's valuable comments for our draft! We have responded the reviewer's comments by clarifying the unclear points and making corresponding revisions for each part of our paper. Given the NeurIPS final discussion deadline (08/09) is approaching, we really hope to have a further discussion with the reviewer vyrt to see if our responses solve the reviewer's concerns.
>
> Thank you very much for your time!
>
> Best wishes,
>
> Authors of Paper1207

---

> > ### Comment · Reviewer_vyrt · 2022-08-08
> > **Post rebuttal**
> >
> > I’d like to thank the authors for the rebuttal. The rebuttal addresses many of my concerns and the authors promise to update and improve the presentation of the paper and outline ways of how they will do so. In terms of evaluation, the authors added interesting experiments in the rebuttal that makes the validation stronger. However, I agree with other reviewers that adding experiments that would go beyond small-scale datasets would strengthen the paper. Moreover, I’m still unconvinced about the potential impact of this work. Nevertheless, after the rebuttal the paper is quite complete with some interesting observations and I'll update my score accordingly.

---

> > > ### Author Response · Authors · 2022-08-09
> > > **The Response to the Reviewer vyrt**
> > >
> > > Dear Reviewer vyrt,
> > >
> > > We appreciate the reviewer's recognition of our rebuttal and pointing out the disadvantages of the current draft. We will further clarify our unclear description and supplement the corresponding content for our final version.
> > >
> > > Thank you very much for your time!
> > >
> > > Best wishes,
> > >
> > > Authors of Paper1207

---

### Meta-Review · Area_Chair_ahR6 · 2022-08-24

**Recommendation:** Accept
**Confidence:** Certain

**Metareview:**

The paper studies the use of random weights together with learnable masks. Authors demonstrate that such training approach for neural network can reduce the model storage requirements and has applications to network compression.

Reviewer appreciated  the novelty of the idea and the extensive experiments on various architectures.  Adding experiments that would go beyond small-scale datasets would further strengthen the quality of the paper and its potential impact.

**Award:**

No

---

### Decision · Program_Chairs · 2022-09-14

Accept